Identification of pre-flexion fish larvae from the western South Atlantic using DNA barcoding and morphological characters

Pozzobon Allan Pierre Bonetti 1 2 allanpbpozzobon@gmail.com
http://orcid.org/0000-0002-9374-8661 Ready Jonathan Stuart 3
Di Dario Fabio 1 2
http://orcid.org/0000-0001-9576-2849 Nunes-da-Fonseca Rodrigo 1 2
1 Instituto de Biodiversidade e Sustentabilidade (NUPEM), Universidade Federal do Rio de Janeiro (UFRJ) , Macaé, Rio de Janeiro , Brazil
2 Programa de Pós-Graduação em Ciências Ambientais e Conservação (PPG-CiAC), Instituto de Biodiversidade e Sustentabilidade (NUPEM), Universidade Federal do Rio de Janeiro (UFRJ) , Macaé, Rio de Janeiro , Brazil
3 Group for Integrated Biological Investigations, Center for Advanced Biodiversity Studies, Federal University of Pará (UFPA) , Belém, Pará , Brazil
Cheung Man Kit
Electronic publication date: 2024 Jul 25
Publication date: 2024
Volume: 12
Electronic Location ID: e17791
Received 2024 Apr 17; Accepted 2024 Jul 1
Copyright: © 2024 Pozzobon et al.
Copyright year: 2024
Copyright holder: Pozzobon et al.
License: This is an open access article distributed under the terms of the Creative Commons Attribution License, which permits unrestricted use, distribution, reproduction and adaptation in any medium and for any purpose provided that it is properly attributed. For attribution, the original author(s), title, publication source (PeerJ) and either DOI or URL of the article must be cited.
License URL: https://creativecommons.org/licenses/by/4.0/

Keywords: Ichthyoplankton, COI, Costal islands

Funding: FAPERJ (Fundação de amparo à pesquisa do estado do Rio de Janeiro) (PDR-10 204.315/2021) CNPq (Conselho Nacional de Desenvolvimento Científico e Tecnológico—PROTAX 443302/2020) This work was supported by FAPERJ (Fundação de amparo à pesquisa do estado do Rio de Janeiro). Financial support to Allan Pierre Bonetti Pozzobon was provided by FAPERJ (PDR-10 204.315/2021). Financial support to Fabio Di Dario was provided by CNPq (Conselho Nacional de Desenvolvimento Científico e Tecnológico—PROTAX 443302/2020). The funders had no role in study design, data collection and analysis, decision to publish, or preparation of the manuscript.

==============================
Knowledge on species composition is the first step necessary for the proper conservation and management of biological resources and ecologically relevant species. High species diversity and a lack of diagnostic characters for some groups can impose difficulties for taxonomic identification through traditional methodologies, and ichthyoplankton (fish larvae and eggs) are a good example of such a scenario. With more than 35.000 valid species of fishes worldwide and overall similar anatomies in early developmental stages in closely related groups, fish larvae are often hard to be identified at the species or even more encompassing taxonomic levels. To overcome this situation, molecular techniques have been applied, with different markers tested over the years. Cytochrome c oxidase I (COI) is the most commonly used marker and now has the broadest public reference libraries, providing consistent results for species identification in different metazoan studies. Here we sequenced the mitochondrial COI-5P fragment of 89 fish larvae collected in the Campos Basin, coastal southeastern Brazil, and compared these sequences with references deposited in public databases to obtain taxonomic identifications. Most specimens identified are species of the Blenniiformes, with Parablennius and Labrisomus the most frequently identified genera. Parablennius included two species (P. marmoreus and P. pilicornis), while Labrisomus included three species (L. cricota, L. conditus and L. nuchipinnis). Anatomy of these molecularly identified specimens were then analyzed with the intention of finding anatomical characters that might be diagnostically informative amongst the early development stage (pre-flexion) larvae. Ventral pigmentation patterns are proposed as useful markers to identify Labrisomus species. However, additional specimens are needed to confirm if the character holds stability through the geographic distribution of the species.

Introduction

Fishes represent a non-monophyletic group that includes more than 35.000 species or about half the recognized species of vertebrates (Nelson, Grande & Wilson, 2016; Facey et al., 2023). Surveying and monitoring such diversity are huge tasks, which are even more challenging considering the breadth of niches occupied and adaptations present across different lineages. Many fish species spawn in the water column and the early stages of development, eggs and fish larvae (which together form the ichthyoplankton), compose a significant part of the planktonic community (Miller & Kendall, 2009; Bonecker et al., 2014; Zhang, Jiang & Li, 2022).

Studies focused on the ichthyoplankton can provide unique information, not only regarding species diversity in a region, but also informing temporo-spatial patterns in the reproductive activity of species. Such data is particularly of interest for the management of fisheries or for the evaluation and monitoring of rare and threatened species and ecosystems. Unfortunately, taxonomic identification of ichthyoplankton at the species level is often hard to achieve using traditional methodologies that generally employ anatomical characters, the majority of which are only informative for adult specimens, with limited diagnostic characters available for early life stages (Richards, 2005; Miller & Kendall, 2009; Reynalte-Tataje et al., 2020; Zhang, Jiang & Li, 2022).

Since the compilation of Moser et al. (1984) on the ontogeny and systematics of fishes, some catalogs focusing on fish larvae have been published, usually targeting the fauna of specific regions of the world. For example, Moser (1996) focused on the California coast, Richards (2005) on the western and central North Atlantic and Bonecker et al. (2014) on the northern Rio de Janeiro State, Brazil. Most characters indicated as useful for species identification on these studies are based on meristic (countable) characters, such as the number of myomeres/vertebrae along the body, and number of fin-rays, which have limited applicability specially if the specimens or parts of their bodies are damaged. In addition, the connection between diagnostic features of adults and larvae of fishes are also not always possible, especially in groups where the two stages have divergent body forms, e.g., Pleuronectiformes (flatfishes) and Anguilliformes (eels and morays). In fish larvae, taxonomic characters tend to be markedly stage and size specific, therefore for species identification, taxonomic descriptions and direct comparisons among specimens of the same species are more frequently used for juveniles and adults (Strauss & Bond, 1990).

In this context, the use of molecular markers has been applied alongside traditional methodologies in integrated taxonomic studies to reevaluate species diversity in several taxa, representing an interesting alternative for species identification in complex cases such as the ichthyoplankton. Hebert et al. (2003), Hebert, Ratnasingham & Dewaard (2003), for instance, proposed a fragment of the mitochondrial gene Cytochrome c Oxidase I (COI-5P) as the molecular marker of choice for distinguishing species within the same genus, relying on differences observed among species of several taxonomic groups. However, the use of molecular markers for diagnostic surveys requires some specificities, as the presence of a “barcoding distance gap”, that refers to the difference between mean intra- and interspecific genetic distances (Lira et al., 2022; Antil et al., 2023), and a reference database of species for known taxa that is as complete as possible.

Since the early proposal of COI-5P as the DNA barcoding fragment (Hebert et al., 2003; Hebert, Ratnasingham & Dewaard, 2003), different projects and consortiums were implemented that focused on specific taxonomic groups. After Ward et al. (2005) implemented COI-5P as a tool for exploring marine fish species diversity in Australia, the FISH-BOL campaign was launched with the overarching goal of producing COI sequences for all world’s fishes, with a minimum of five specimens per species. Ward (2012) published a review of the achievements of the FISH-BOL campaign and, at that time, around 8,000 of the 31,000 recognized species of fishes had their COI-5P fragment sequenced, comprising a valuable public reference library of COI sequences for the group.

In 2010 the International Barcode of Life project was launched (iBOL–ibol.org), with the aim to centralize the different initiatives and consortia that had previously been established. Since 2014, the Barcode of the Life Data System (BOLD) stores and shares COI-5P and other reference sequences, linking different databases and laboratories. Today, BOLD (boldsystems.org) holds sequences of more than 250,000 animal species, more than 72,000 plant species, and more than 25,000 fungi species (boldsystems.org, Ratnasingham & Hebert, 2007). Specifically for the Actinopterygii or ray-finned fishes, more than 22,000 species have reference sequences (boldsystems.org; Ratnasingham & Hebert, 2007). The large number of COI-5P sequences available for species of fishes and the presence of a “barcoding gap” for most of them (98%; Ward, Hanner & Hebert, 2009), make the COI-5P fragment the optimal option when the goal is to identify fish species using molecular markers.

Despite these promising new technologies, molecular species identification should be performed with caution because the “barcoding gap” is not universal among the different species or taxonomic groups, and there may be a lack of reference sequences for comparison, especially for new and less used molecular markers (Lira et al., 2022; Antil et al., 2023). Furthermore, the use of anatomical or morphological characters is more economically inclusive, as financial resources are scarce worldwide, with limited access to equipment and/or technology to conduct surveys using molecular tools (Stein et al., 2014). Therefore, despite the development in the last years and the reliability of molecular identification, it is still very important to search for diagnostic morphological characters especially for early life stages of fishes. This is even more important in the case of developing regions, which often harbor a significant portion of the world’s (often unrecognized) biodiversity and are under significant pressure from expanding human activities.

The Campos Basin, southeastern Brazil, is an example of such a region where human endeavor is prominent, as the basin is one of the most important regions for offshore exploration of oil and gas worldwide (Poubel & Junior, 2017). Despite the economic relevance of the region, its marine biological diversity is still insufficiently known (e.g., Mincarone et al., 2016), hindering efforts towards the proper management and conservation of the marine ecosystems of the region. In this context, our goal is to present the first study aimed at the identification of early stages of development (pre-flexion) of fish larvae collected from the Campos Basin, southeastern Brazil, using the COI-5P molecular marker. After assigning samples to species, we then explored anatomical variation among larvae with the intention of identifying characters that are potentially useful for species identification.

Materials and Methods

Sampling sites and specimen collection

Ichthyoplankton were sampled using twin bongo nets (mesh sizes 300 and 500 µm) towed at 0.5 m depth from surface for 10 min (Kelso & Rutherford, 1996; Castro, Bonecker & Valentin, 2005; Food Agriculture Organization (FAO), 2017) in coastal waters near islands from the northeastern part of Rio de Janeiro State, in the Campos Basin, Brazil. Two collecting expeditions were conducted, one in March and a second in April 2022 (Fig. 1). The first expedition was aimed at collecting around the Santana Archipelago, off the city of Macaé (22°24′37.7″S/41°42′20.7″W), in six different sampling localities. In the second expedition, the same localities around the Santana Archipelago were again sampled using the same protocol, with additional collecting at Feia Island, another coastal island off the city of Armação dos Búzios (22°43′26.7″S/41°55′18.7″W). Towed material was directed to the bottom of bongo nets, where a hollow plastic cup accumulated the collected material. The cup was then detached and all the collected material was transferred to pots where samples were preserved in 95% ethanol and transported to be stored in freezers for later sorting of fish larvae and eggs using stereomicroscopes in the laboratories of the Instituto de Biodiversidade e Sustentabilidade, Universidade Federal do Rio de Janeiro (NUPEM/UFRJ).

Figure 1 Study area in the northern portion of Rio de Janeiro State, Campos Basin, southeastern Brazil, between the cities of Macaé (north) and Armação dos Búzios (south).

Red dots represent collection localities around the Santana Archipelago and Feia Island. Brazilian and RJ state maps, shape file from IBGE: https://portaldemapas.ibge.gov.br. Map data: Google, ©2024 Earthstar Geographics SIO.

Molecular techniques

In the laboratory, fish larvae and eggs were separated from the remaining plankton. Eighty-nine randomly selected individual fish larvae were then photographed with a high-resolution stereoscope (LEICA M205 FA) at the Unidade Integrada de Imagem NUPEM/UFRJ, to get high quality images for posterior anatomical investigation. After specimens were photographed, the WIZARD PROMEGA kit was used to perform DNA extraction, with individuals digested entirely, due to prevalence of small fish larvae (total length <5 mm). The molecular marker COI-5P was amplified through Polymerase Chain Reaction (PCR) using the primers of Ward et al. (2005). Thermal cycling conditions consisted of an initial denaturation step of 2 min at 95 °C, followed by 35 cycles of denaturation (94 °C, 45 s), annealing (53 °C, 30 s), and extension (72 °C, 45 s), with a final extension at 72 °C for 8 min, before being held at 8 °C at the end of reaction. Electrophoresis of the PCR products was conducted in a 1.5% agarose gel to confirm successful amplification, and sequencing of the PCR products was performed by the commercial services of Macrogen Ltd. (South Korea) with the same primers used in PCR reaction.

Forward and reverse sequences were aligned using the DAMBE software (Xia & Xie, 2001), resulting in the full-length COI-5P fragment (~654 bp) for each specimen. The closest molecular taxonomic identification was then determined by comparing these sequences to the reference databases of NCBI (GenBank), using the nBLAST algorithm, and BOLD Systems, using the Identification Engine (hereafter called BOLD IE). For NCBI nBLAST, high similarity (>98%) between sequences with good coverage (overlap between sequences—>95%) are indicative that sequences belong to the same species (Hebert et al., 2003; Hebert, Ratnasingham & Dewaard, 2003). For the BOLD IE, identification can be assessed both based on sequence identity and through the placement of the submitted sequence within a local tree.

Additional checking of taxonomic identification of sequences was performed through phylogenetic analyses and K2P intra/inter specific genetic distances for larvae sequences with different results for species identification on the two databases tested (NCBI and BOLD) with close percentages of similarity between NCBI and BOLD or if low percentages of similarity were observed (<85%). Phylogenetic analyses and K2P intra/inter specific genetic distances were performed in MEGA X (Kumar et al., 2018).

Since most uncertainty concerning larvae identification was found in the Labrisomidae (see Results), we used the phylogeny of the Blenniiformes proposed by Lin & Hastings (2013) to further assess the identification of sequences of the family. In order to do that, maximum likelihood (ML) and maximum parsimony (MP) were performed, and bootstrap analyses (Felsenstein, 1985) were implemented to produce topologies and give statistical support to nodes, using 1.000 pseudoreplicates. A total of 65 reference sequences, representing 23 species of the ingroup and one outgroup (Opistognathus aurifrons, Opistognathidae), in addition to 39 sequences from our larvae dataset that could not be confidently identified by nBLAST and BOLD IE (total of 104 sequences), were included in both ML and MP analyses.

Taxonomic classification above the genus level follow Ghezelayagh et al. (2022) and Near & Thacker (2024).

Anatomical counts and observations

After monophyletic lineages were determined and likely species identified, high-resolution images of larval specimens were grouped based on the molecular identifications. Anatomical characters were then explored for genera containing more than five specimens identified, in order to increase the possibility of detecting anatomical variation which would not be detected if based on a fewer number of specimens. To assess possible diagnostic anatomical features, pigmentation patterns, including the number and placement of individual melanophores, as well as the shape and size of pigmented patches, frequently recognized among the most taxonomically useful characters for discriminating closely related species, were investigated (Moser et al., 1984; Strauss & Bond, 1990; Moser, 1996; Richards, 2005).

Results

In the first expedition, 141 fish larvae and 21.514 eggs were collected around the Santana Archipelago. The second expedition resulted in the collection of 114 fish larvae and 5.541 eggs, from around both the Santana Archipelago and Feia Island, resulting in a total sample set of 255 fish larvae and 27.055 eggs (Table 1).

Table 1 Number of fish eggs and larvae collected in each collection, separated by net mesh sizes used during sampling.

Collection
event	Geographic
location	Sampling
date	Eggs
500 µm	Eggs
300 µm	Larvae
500 µm	Larvae
300 µm	
AR01	Santana Archipelago	03/2022	2,354	4,754	6	49	
AR02	Santana Archipelago	03/2022	3,389	1,926	1	13	
AR03	Santana Archipelago	03/2022	1,134	858	2	31	
AR04	Santana Archipelago	03/2022	1,023	835	2	16	
AR05	Santana Archipelago	03/2022	543	434	2	4	
AR06	Santana Archipelago	03/2022	2,478	1,786	1	14	
BZ1	Feia Island	04/2022	71	165	10	18	
BZ2	Feia Island	04/2022	99	171	6	4	
BZ3	Feia Island	04/2022	137	371	10	33	
BZ4	Feia Island	04/2022	97	208	4	2	
BZ5	Feia Island	04/2022	98	98	0	2	
BZ6	Feia Island	04/2022	74	135	1	5	
MC1	Santana Archipelago	04/2022	20	116	1	4	
MC2	Santana Archipelago	04/2022	424	447	0	4	
MC3	Santana Archipelago	04/2022	1,004	1,253	0	2	
MC4	Santana Archipelago	04/2022	24	174	1	0	
MC5	Santana Archipelago	04/2022	157	160	1	0	
MC6	Santana Archipelago	04/2022	8	30	0	6	

Of the eighty-nine fish larvae selected for photographs and molecular analyses, 80 were at an early stage of maturation, identified as the pre-flexion stage of development, ranging from <1.5 mm to a maximum of <5 mm total length. Nine fish larvae with more than 5 mm in total length were identified at the post-flexion stage of development. Between two and four images were taken from each specimen, resulting in more than 250 high resolution images from fish larvae.

Molecular identification

Among the COI-5P sequences produced for the 89 specimens submitted for NCBI nBLAST and the BOLD IE analyses, more than 84% (75 specimens) showed identifications with similarities above 96%, providing identity of specimens with a high degree of confidence at least to the genus level. Among these, 68 specimens (76.4%) presented similarity above 98%, as expected for individuals belonging to the same species. A total of 20 species were identified representing six orders (Blenniiformes, Acanthuriformes, Syngnathiformes, Carangiformes, Clupeiformes and Gobiiformes). More than 80% of the larvae identified were members of the Blenniiformes (75 specimens), whereas the Gobiiformes, and Clupeiformes were each represented by a single specimen. The Blenniidae and Labrisomidae were the most frequently identified families, with most specimens identified in the genera Parablennius and Labrisomus. Detailed information on the percentage of similarity for each specimen in relation to available reference sequences and Genbank accession numbers for the new sequences are presented in Table 2.

Table 2 Taxonomic identification of fish larvae, percentage of similarity with the closest match in each database.

Different taxonomic identification between reference databases is represented by “*”.

Larvae ID	Genbank accesion	Order	Family	Species nBLAST/BOLD IE	ID%-nBLAST	ID%-BOLD IE	
AR06-L1*	PP578103	Blenniiformes	Labrisomidae	Labrisomus sp./Labrisomus conditus	98.17%	99.84%	
AR06-L2	PP578104	Blenniiformes	Blenniidae	Scartella cristata	96.51%	97.06%	
AR06-L3	PP578105	Blenniiformes	Blenniidae	Parablennius marmoreus	97.33%	97.35%	
AR06-L4	PP578106	Blenniiformes	Blenniidae	Scartella cristata	99.17%	99.25%	
AR06-L5*	PP578107	Blenniiformes	Labrisomidae	Labrisomus sp./Labrisomus conditus	97.69%	98.08%	
AR05-L1	PP578108	Blenniiformes	Blenniidae	Scartella cristata	99.66%	99.66%	
AR05-L2*	PP578109	Blenniiformes	Labrisomidae	Labrisomus sp./Labrisomus conditus	98.32%	99.85%	
AR04-L1	PP578110	Blenniiformes	Blenniidae	Parablennius marmoreus	97.86%	98.14%	
AR04-L2	PP578111	Blenniiformes	Blenniidae	Parablennius pilicornis	98.77%	99.00%	
AR04-L3	PP578112	Blenniiformes	Blenniidae	Hypleurochilus fissicornis	100%	100%	
AR04-L5	PP578113	Blenniiformes	Labrisomidae	Malacoctenus delalandii	99.65%	99.65%	
AR03-L1	PP578114	Blenniiformes	Blenniidae	Parablennius marmoreus	97.69%	97.66%	
AR03-L2	PP578115	Blenniiformes	Blenniidae	Parablennius pilicornis	98.92%	99.00%	
AR03-L3	PP578116	Blenniiformes	Blenniidae	Parablennius marmoreus	96.93%	98.14%	
AR03-L4	PP578117	Blenniiformes	Blenniidae	Parablennius marmoreus	98.22%	98.21%	
AR03-L5	PP578118	Blenniiformes	Blenniidae	Parablennius marmoreus	98.22%	98.21%	
AR03-L6	PP578119	Blenniiformes	Blenniidae	Parablennius marmoreus	98.32%	98.31%	
AR03-L7	PP578120	Blenniiformes	Blenniidae	Parablennius marmoreus	97.67%	97.85%	
AR02-L1	PP578121	Blenniiformes	Blenniidae	Parablennius marmoreus	98.32%	98.31%	
AR02-L2	PP578122	Blenniiformes	Blenniidae	Parablennius marmoreus	98.32%	98.31%	
AR02-L3	PP578123	Blenniiformes	Blenniidae	Hypleurochilus fissicornis	100%	100%	
AR01-L1	PP578124	Blenniiformes	Blenniidae	Parablennius marmoreus	98.17%	98.31%	
AR01-L2	PP578125	Blenniiformes	Dactyloscopidae	Dactyloscopus foraminosus	99.23%	99.23%	
BZ3-L1*	PP578126	Blenniiformes	Labrisomidae	Paraclinus sp./Paraclinus fasciatus	84.80%	84.43%	
BZ3-L2*	PP578127	Blenniiformes	Labrisomidae	Labrisomus sp./Labrisomus conditus	97.40%	97.98%	
BZ3-L3*	PP578128	Blenniiformes	Labrisomidae	Labrisomus nuchipinnis/Labrisomus cricota	99.69%	99.85%	
BZ3-L4*	PP578129	Blenniiformes	Labrisomidae	Paraclinus sp./Paraclinus fasciatus	84.66%	84.43%	
BZ3-L5*	PP578130	Blenniiformes	Labrisomidae	Labrisomus sp./Labrisomus conditus	98.32%	99.85%	
BZ3-L6*	PP578131	Blenniiformes	Labrisomidae	Paraclinus sp./Paraclinus fasciatus	84.80%	84.43%	
BZ3-L7	PP578132	Blenniiformes	Dactyloscopidae	Dactyloscopus foraminosus	99.39%	99.39%	
BZ6-L1*	PP578133	Blenniiformes	Labrisomidae	Labrisomus sp./Labrisomus cricota	99.35%	99.84%	
BZ6-L2*	PP578134	Blenniiformes	Labrisomidae	Labrisomus nuchipinnis/Labrisomus cricota	99.03%	99.81%	
BZ6-L4*	PP578135	Blenniiformes	Labrisomidae	Labrisomus sp./Labrisomus conditus	98.32%	99.85%	
BZ6-L5*	PP578136	Syngnathiformes	Syngnathidae	Microphis aculeatus/Microphis lineatus	93.74%	99.85%	
BZ6-L6	PP578137	Clupeiformes	Clupeidae	Opisthonema oglinum	98.32%	98.46%	
AR01-L1s2	PP578138	Blenniiformes	Dactyloscopidae	Dactyloscopus foraminosus	99.11%	99.1%	
AR01-L2s2	PP578139	Blenniiformes	Blenniidae	Parablennius marmoreus	98.32%	98.31%	
AR01-L3s2	PP578140	Blenniiformes	Blenniidae	Parablennius pilicornis	98.92%	99%	
AR04-L1s2	PP578141	Blenniiformes	Blenniidae	Parablennius marmoreus	98.32%	98.31%	
AR04-L2s2	PP578142	Blenniiformes	Blenniidae	Parablennius marmoreus	98.32%	98.31%	
AR04-L3s2*	PP578143	Blenniiformes	Blenniidae	Scorpaenopsis venosa/Hypsoblennius invemar	84%	100%	
AR04-L4s2*	PP578144	Blenniiformes	Labrisomidae	Paraclinus sp./Paraclinus fasciatus	84.80%	84.43%	
BZ1-L1	PP578145	Acanthuriformes	Gerreidae	Eucinostomus argenteus	99.85%	99.85%	
BZ1-L2	PP578146	Acanthuriformes	Gerreidae	Eucinostomus argenteus	99.69%	99.69%	
BZ1-L3	PP578147	Acanthuriformes	Gerreidae	Eucinostomus argenteus	99.39%	99.51%	
BZ1-L4	PP578148	Acanthuriformes	Gerreidae	Eucinostomus argenteus	100%	100%	
BZ1-L6	PP578149	Acanthuriformes	Gerreidae	Eucinostomus argenteus	100%	100%	
BZ1-L7	PP578150	Acanthuriformes	Gerreidae	Eucinostomus argenteus	98.62%	99.02%	
BZ1-L8*	PP578151	Blenniiformes	Labrisomidae	Paraclinus sp./Paraclinus fasciatus	84.66%	84.43%	
BZ1-L9	PP578152	Blenniiformes	Blenniidae	Parablennius marmoreus	98.32%	98.31%	
BZ1-L10	PP578153	Acanthuriformes	Sciaenidae	Stellifer rastrifer	99.85%	99.85%	
BZ2-L1	PP578154	Acanthuriformes	Gerreidae	Eucinostomus argenteus	100%	100%	
BZ2-L2*	PP578155	Blenniiformes	Labrisomidae	Labrisomus sp./Labrisomus conditus	98.17%	99.39%	
BZ4-L1*	PP578156	Syngnathiformes	Syngnathidae	Microphis aculeatus/Microphis lineatus	94.05%	100%	
BZ4-L2*	PP578157	Blenniiformes	Labrisomidae	Labrisomus nuchipinnis/Labrisomus cricota	99.23%	99.42%	
BZ4-L3*	PP578158	Blenniiformes	Labrisomidae	Labrisomus nuchipinnis/Labrisomus cricota	99.23%	99.81%	
BZ4-L4*	PP578159	Blenniiformes	Labrisomidae	Paraclinus sp./Paraclinus fasciatus	84.80%	84.43%	
BZ4-L5*	PP578160	Blenniiformes	Labrisomidae	Labrisomus sp./Labrisomus conditus	98.45%	99.39%	
BZ4-L6	PP578161	Gobiiformes	Gobiidae	Evorthodus lyricus	96.64%	96.62%	
BZ5-L1	PP578162	Blenniiformes	Pomacentridae	Abudefduf saxatilis	100%	100%	
BZ5-L2	PP578163	Blenniiformes	Pomacentridae	Stegastes fuscus	99.84%	100%	
BZ6-L3	PP578164	Carangiformes	Carangidae	Oligoplites saurus	99.08%	99.08%	
BZ1-L5	PP578165	Blenniiformes	Blenniidae	Parablennius marmoreus	97.71%	97.84%	
BZ2-L3*	PP578166	Blenniiformes	Labrisomidae	Paraclinus sp./Paraclinus fasciatus	84.80%	84.43%	
BZ2-L4*	PP578167	Blenniiformes	Labrisomidae	Labrisomus sp./Labrisomus conditus	98.65%	99.42%	
BZ2-L5*	PP578168	Blenniiformes	Labrisomidae	Paraclinus sp./Paraclinus fasciatus	84.80%	84.43%	
BZ2-L6*	PP578169	Blenniiformes	Labrisomidae	Paraclinus sp./Paraclinus fasciatus	84.82%	84.59%	
BZ2-L7	PP578170	Blenniiformes	Blenniidae	Parablennius marmoreus	98.32%	98.31%	
BZ2-L10*	PP578171	Blenniiformes	Labrisomidae	Labrisomus nuchipinnis/Labrisomus cricota	99.69%	99.85%	
BZ3-L8*	PP578172	Blenniiformes	Labrisomidae	Labrisomus nuchipinnis/Labrisomus cricota	99.85%	100%	
BZ3-L9*	PP578173	Blenniiformes	Labrisomidae	Labrisomus sp./Labrisomus nuchipinnis	99.41%	100%	
BZ3-L11*	PP578174	Blenniiformes	Labrisomidae	Labrisomus nuchipinnis/Labrisomus cricota	99.69%	99.85%	
BZ3-L12	PP578175	Blenniiformes	Labrisomidae	Malacoctenus delalandii	99.26%	99.26%	
MC1-L1*	PP578176	Blenniiformes	Labrisomidae	Paraclinus sp./Paraclinus fasciatus	84.80%	84.43%	
MC1-L2*	PP578177	Blenniiformes	Labrisomidae	Labrisomus sp./Labrisomus nuchipinnis	99.11%	99.69%	
MC1-L3*	PP578178	Blenniiformes	Labrisomidae	Labrisomus sp./Labrisomus conditus	98.02%	99.23%	
MC1-L4*	PP578179	Blenniiformes	Labrisomidae	Labrisomus sp./Labrisomus conditus	98.17%	99.39%	
MC1-L5	PP578180	Blenniiformes	Labrisomidae	Malacoctenus delalandii	99.63%	99.81%	
MC2-L1	PP578181	Blenniiformes	Labrisomidae	Malacoctenus brunoi	99.53%	99.53%	
MC2-L2*	PP578182	Blenniiformes	Labrisomidae	Paraclinus sp./Paraclinus fasciatus	84.82%	84.59%	
MC2-L3	PP578183	Blenniiformes	Dactyloscopidae	Dactyloscopus foraminosus	99.39%	99.39%	
MC3-L1	PP578184	Blenniiformes	Pomacentridae	Abudefduf saxatilis	99.27%	100%	
MC4-L1	PP578185	Carangiformes	Carangidae	Chloroscombrus chrysurus	97.67%	99.83%	
MC5-L1	PP578186	Blenniiformes	Dactyloscopidae	Dactyloscopus foraminosus	99.23%	99.23%	
MC6-L1*	PP578187	Blenniiformes	Labrisomidae	Paraclinus sp./Paraclinus fasciatus	84.80%	84.43%	
MC6-L2*	PP578188	Blenniiformes	Labrisomidae	Paraclinus sp./Paraclinus fasciatus	84.66%	84.43%	
MC6-L3*	PP578189	Blenniiformes	Labrisomidae	Paraclinus sp./Paraclinus fasciatus	84.82%	84.59%	
MC6-L4	PP578190	Blenniiformes	Blenniidae	Parablennius marmoreus	97.96%	98.21%	
MC6-L5	PP578191	Blenniiformes	Blenniidae	Hypleurochilus fissicornis	99.34%	100%	

Thirty-nine sequences had different species identifications from the two databases or had low values of similarity for species identification in both databases. These sequences were phylogenetically analyzed using two different methodologies (ML/MP). Both methods resulted in the same monophyletic groups with strong bootstrap support (>98%, except for P. fasciatus, 73% in MP). Each main clade only contained individuals from a single species, reinforcing that the clades where larval sequences are clustered represent reliable taxonomic identifications (Fig. 2).

Figure 2 ML phylogenetic tree of the Labrisomidae.

Numbers on branches represent bootstrap values (ML/MP), an asterisk (*) represents ≥99% for ML and MP, a dash (-) represents branches not recovered by the method. Species are highlighted in bold if they represent fish larvae specimens identified in the present study (new sequences underlined in the tree). Scale indicates genetic divergence.

The genera with most species and specimens identified were Labrisomus (L. nuchipinnis, L. cricota and L. conditus, with 21 specimens) and Parablennius (P. marmoreus and P. pilicornis, also with 21 specimens), which represent all species of those genera currently reported for the western South Atlantic (Menezes et al., 2003; Levy et al., 2013; Lin & Hastings, 2013; Fricke, Eschmeyer & Van Der Laan, 2023). Eighteen larval sequences have a high degree of similarity (96.7–98.3%) with P. marmoreus from the western North Atlantic (Florida and Caribbean) deposited in the databases used as references in our study. Three further sequences are closely related to P. pilicornis, again with a high degree of similarity (>98%).

In the case of Labrisomus, nBLAST and BOLD IE species identifications are conflicting (Table 2), but the phylogenetic analyses agree that our sequences represent L. conditus (11 sequences), L. cricota (eight sequences,), and L. nuchipinnis (two sequences), with a high support for all clades (bootstrap = 100%, except for L. cricota, with 99% in MP; Fig. 2). Intra and interspecific genetic distances (K2P) for Labrisomus species, support taxonomic identification, presenting a major gap between the averages observed (0.4% and 17.7%, respectively) (Tables S1 and S2). Other species of the Labrisomidae were identified, including Malacoctenus delalandii (three sequences), both according to nBLAST and BOLD IE (>99% similarity for both) and the phylogenetic analyses (100% bootstrap). An additional sequence was identified as Malacoctenus brunoi, again based on both nBLAST and BOLD IE (>99% similarity for both) and the phylogenetic analyses (100% bootstrap). Three further sequences were assigned to species of the Pomacentridae, with two species identified and with high values of similarity for both (>99%)

Identification of those sequences at the species level is therefore highly likely, but in other cases taxonomic precision was not achieved based on the methods employed and comparative sequences included in our study. Fourteen sequences identified as related to Paraclinus have a lower degree of similarity according to nBLAST and BOLD IE (84.43–84.82%). Our phylogenetic analyses also failed to recover Paraclinus as monophyletic, but the 14 new sequences form a distinct clade (100% bootstrap) that is included in a larger clade containing other species of Paraclinus (P. sini and P. asper) with relatively low support (bootstrap < 70%; Fig. 2). A great gap is also observed between averages of intra and interspecific genetic distances for Paraclinus species (1.9% and 21%, respectively) (Tables S1 and S2). However, intraspecific divergence of P. nigripinnis and P. fasciatus are higher (3.9% and 9.7%, respectively) than the ones observed for the remaining species of Paraclinus (0.8% average) (Table S1).

The Acanthuriformes was the order with the second highest number of specimens identified in our analyses. Seven specimens were identified as Eucinostomus argenteus (Gerreidae), with high values of similarity (98.6–100%) with comparative sequences. A single specimen belonging to the Sciaenidae was also identified, again with high values of similarity with sequences from both databases (>99%). Two specimens were identified as a single species of the Syngnathiformes, and two other specimens were identified as distinct species of the Carangiformes, with high values of similarity in at least one of the databases used for analysis (>98%). Only one specimen was assigned to each of the orders Clupeiformes and Gobiiformes, but with differing degrees of similarity (Table 2). In the Clupeiformes, our specimen can be reliably identified as Opisthonema oglinum (>98%), but in the case of the specimen identified as a member of the Gobiiformes a relatively lower level of similarity was recovered (>96%). Therefore, in this case identification was confident only at the genus level (Evorthodus), with only E. lyricus currently reported for the region (Table 2).

Morphology

No visible pigmentation pattern with taxonomic diagnostic applicability was found among the larvae of the two Parablennius species identified here (Fig. 3). The number and form of ventral spots do not reliably distinguish the two species (P. marmoreus = 20–29/P. pilicornis = 22–23). Parablennius pilicornis seems to have a lower number of ventral spots, but this variation might be due to the low number of specimens examined (3–Table 3).

Figure 3 Fish larvae molecularly identified as Parablennius.

Specimens on the left (A to C) were identified as P. pilicornis, and those on the right (D to F) as P. marmoreus. Scale bars = 1 mm.

Table 3 Morphological characters identified for fish larvae for which the number of specimens identified (in parentheses) in this study was equal or greater than five for the genus.

Molecular identification	Total number of ventral spots	Pre-anal spots	Post-anal spots	Spot morphology	
Parablennius marmoreus (18)	20–29	0–01	20–26	Mostly irregular	
Parablennius pilicornis (03)	22–23	0	22–23	Mostly irregular	
Labrisomus conditus (11)	02–06	01–03	01–03	Variable	
Labrisomus cricota (08)	07–11	02–04	05–09	Variable	
Labrisomus nuchipinnis (02)	06–07	0–01	06	Variable	
Eucinostomus argenteus (07)	08–11	03	05–08	Mostly regular circles	

Different species of Labrisomus, in turn, could be reliably identified based on different numbers and relative size of the ventral spots. Specimens of Labrisomus conditus have a lower total number of ventral spots (02–06), L. nuchipinnis has an intermediate number (06-07), and L. cricota has the highest number of spots (07-11) (Table 3). Pre-anal and post-anal spot counts in particular seem to be more effective in terms of specimen identification. In L. conditus, one to three post-anal spots were identified, whereas both L. nuchipinnis and L. cricota have five or more post-anal spots (Table 3). Labrisomus conditus also has a relatively large spot on the ventral mid-section of the body, contrasting with the mostly similarly sized spots of L. cricota and L. nuchipinnis at this region of the body (Fig. 4). Labrisomus cricota, in turn, has two or more pre-anal spots, whereas L. nuchipinnis has one pre-anal spot. Overall, considering spot size and shape, pigmentation pattern of L. nuchipinnis seems to be more similar to that present in L. cricota than in relation to L. conditus, where spots along the ventral part of the body are more uniform (Fig. 4).

Figure 4 Fish larvae molecularly identified as Labrisomus.

Specimens (A and B) were identified as L. conditus; specimens (C and D) identified as L. cricota; (E and F) were identified as L. nuchipinnis. Arrows indicated large spot on the ventral mid-section of L. conditus. Scale bars = 1 mm.

A pattern of ventral spots was identified for the E. argenteus larvae. These large larvae (>5 mm) all present well delimited spots compared to the starred or blurred spots observed in the Blenniiformes described above (Fig. 5). The total number of ventral spots ranges from 08 to 11 and there is a constant number of pre-anal spots (03–Table 3).

Figure 5 Fish larvae molecularly identified as Eucinostomus argenteus.

Scale bars = 1 mm.

Discussion

Diversity found

Overall, molecular identification of fish larvae was successful, despite limited sampling year periods that limits our power of inference and discussion. The dominance of cryptobenthic reef species (mostly species of the Blenniiformes and Gobiidae) was expected because the collected sites are near islands surrounded by reef formations. Brandl et al. (2019) observed a similar pattern, with a median of two-third (65.7%) of fish larvae belonging to these taxa near reefs (<10 km) worldwide. Specifically in the western Atlantic, Brandl et al. (2019) recorded a smaller proportion of cryptobenthic species among fish larvae (<60%) in comparison with other parts of the world. However, their study was mostly focused on the Caribbean Sea and the Gulf of Mexico in the Florida Peninsula, which represent relatively enclosed portions of the ocean in terms of circulation of water masses when compared to the region of the western South Atlantic where our samples were collected. It is possible that this condition results in an aggregation of larvae of usually larger and pelagic fishes near islands of the Caribbean and the Gulf of Mexico, likely explaining the differences observed between the results of Brandl et al. (2019) and our study, where more than 80% of identified species belong to cryptobenthic reef fishes.

Recognizing the different species composition of the cryptobenthic fish community is of great concern since this group has been highlighted as having a central role in “Darwin’s paradox”. This paradox questions how coral reefs can maintain such high diversity and productivity in oligotrophic tropical oceans. Brandl et al. (2019) proposed that the cryptobenthic species promote internal reef fish biomass production, accounting for around two-thirds of reef fish larvae and producing around 60% of consumed reef fish biomass. Thus, precise knowledge about the composition of cryptobenthic fish species is fundamental for understanding the maintenance and conservation of reef ecosystems.

Distinguishing between closely related fish species can be challenging, sometimes even when dealing with adult specimens, and the task is even more complicated for ichthyoplankton, resulting in an early use of molecular techniques for identification of these specimens. Pegg et al. (2006) explored COI-5P as a marker for fish larval identification in the Great Barrier Reef in Australia and since then there has been an exponential growth of studies using molecular techniques to identify ichthyoplankton, from six publications between 2000 and 2010, to 75 in the next decade (Lira et al., 2022). Fish larvae from various other regions of the world have been identified by sequencing their COI-5P fragment. Hubert et al. (2010), for instance, focused on 22 species of the Acanthuridae and 16 of the Holocentridae from French Polynesia (Pacific Ocean), highlighting the differences between intra and interspecific genetic divergences, with 20- to 87-fold higher interspecific divergence between congeners compared to intraspecific divergence, also supporting the reliability of molecular taxonomic identifications for fish larvae.

Similarly, Azmir et al. (2020) analyzed fish larvae from Malaysia and found 41 species among the 153 larvae sequenced, also with a great difference between intraspecific average K2P distance (1%) and interspecific divergence (20%). Despite the “gap” observed in their study, five specimens were not identifiable with precision, with low values of identity and similarity for the two databases (GenBank and BOLD—<90%). In Hawaiian waters 92 fish larvae were identified using the COI-5P marker with a mean intraspecific divergence of 0.72% vs. a mean interspecific but congeneric divergence of 25.9%, but even with these genetic divergences, 52 specimens could only be assigned to the genus level because of the low values of identity observed when comparing query sequences with the reference database (Xing et al., 2022). The difference between intra and interspecific genetic distances observed here for Labrisomus species, support the phylogenetic position and taxonomic identification of larvae sequences of this genus. However, even with the great difference between intraspecific average K2P distance (1.9%) and interspecific divergence (21.9%), we were not able to reach a conclusion regarding Paraclinus larvae species in query. These cases of genus level identification reflect the importance of using the most complete reference database available and, if necessary, completing this reference database using identified adult specimens for the sampling region.

In Brazil, marine ichthyoplankton species surveys have been conducted in the north, northeast, and southeast to south regions of the country (Bonecker & Castro, 2018; Katsugawara et al., 2011; Bonecker et al., 2014; Rutkowski et al., 2011). However, none of these studies applied molecular techniques for species identification. Only two studies so far used molecular identification of Brazilian marine ichthyoplankton. Rodrigues et al. (2017) focused on billfish larvae and eggs (Istiophoridae and Xiphiidae) from the southeast region, and this is one of the few studies before ours that also provided photographs of the specimens identified with molecular tools. In their case, images were used to identify larvae of the target families based on prominent characters (four head spines, turned backwards) to select samples for the molecular analyses. However, several specimens, after molecular procedures using COI-5P as marker, were then later identified as Dactylopterus volitans (Dactylopteridae), exemplifying how difficult it is to properly identify fish larvae based on morphology exclusively (Rodrigues et al., 2017). In comparison, Costa et al. (2023) analyzed eggs from lower latitudes, with limited success in the identification at species level, a fact they believe is associated to the lack of reliable reference sequences for the fish fauna of the region.

Limitations of identification

Ko et al. (2013) tested how pronounced misidentification of ichthyoplankton could be, asking five different laboratories to identify fish larvae through morphology and then subsequently conducting molecular procedures. The average accuracy of the identifications, for the five laboratories, were ~75% for genus level and ~43% for species level. Families with commercially targeted species (Scombridae and Serranidae) were among the most misidentified. However, Mene maculata and Microcanthus strigatus were correctly identified by all the laboratories due to their distinctive morphology. Low taxonomic accuracy percentages in ichthyoplankton species identification (30% at genus level) have also been observed when comparing morphological and molecular approaches by Azmir et al. (2017). Mateos-Rivera et al. (2020), in turn, were able to identify almost all fish larvae in their study using anatomical characters, except for seven larvae of the Callionymidae that were identified through DNA barcoding.

Despite good validation of molecular techniques for species identification of ichthyoplankton in the last decade, all these studies stress that the reliability of this approach depends on representative databases with correct taxonomic identifications of the reference data (Ko et al., 2013; Azmir et al., 2017; Mateos-Rivera et al., 2020; Xing et al., 2022). Using two databases to identify species in the similarity analyses in the present study (NCBI and BOLD) helped to achieve reliable identification for almost all specimens (68 specimens presenting >98% of similarity). The four sequences of P. marmoreus with similarities between 97% and 98% were identified based on sequences from the North Atlantic, a situation that might help explaining the relatively lower levels of similarity detected (Table 2). Species molecular identification reliability can be associated with the number of sequences available, for each species, in databases consulted. In the same way, sequences from individuals collected along all the geographic distribution of each species, also improved trustfulness on species identification and the results found here for P. marmoreus are an example of what happen with a lack of a broad coverage of the species distribution and highlights the importance of a continuous improvement of databases.

Specimens with conflicting species identifications were revised using phylogenetic analyses. This helped to confirm the species of Labrisomus, where the different larval sequences could be assigned to species (clear monophyletic clades with high values of bootstrap support; Fig. 2). Larval sequences associated with the genus Paraclinus were not recovered in a monophyletic group with any species amongst the available reference sequences in our analyses, limiting the species level identification of the fourteen larvae sequences we produced within this genus. Incomplete reference sequences in databases are the likely reason for the lack of species definition for these fourteen larvae, since only one of the three species of Paraclinus recognized in the Brazilian coast (Guimarães & Bacellar, 2002) were included in the phylogenetic analyses (P. spectator). The remaining two (P. arcanus and P. rubicundus) were not found in the different databases explored here (no public COI-5P sequences from these species were found). However, the possibility of a cryptic fourth species for Brazilian waters cannot be excluded and the intraspecific divergence observed for P. fasciatus and P. nigripinnis, are evidence that the genus might need a taxonomic revision, where cryptic species can be uncovered and described.

Lack of monophyly for the genus Paraclinus might be due to the fast evolutionary rate of COI-5P and an ancient origin for the genus. Paraclinus has been proposed as one of the first lineages to diverge in the Labrisomidae based on a study that included nuclear molecular markers as well as COI-5P data (Lin & Hastings, 2013). Although COI-5P is probably the molecular marker with the broadest public species sequences libraries, in places with still unexplored species diversity, such as the western South Atlantic, more investment is needed for development of reliable reference sequences libraries that will, in turn, allow more robust molecular species identification analyses.

Morphological characters

Complementary to the development of more broad and inclusive reference sequences databases, the recognition of taxonomic diagnostic anatomical features in fish larvae is of great interest. Here, ventral pigmentation pattern was identified as useful to distinguish among Labrisomus species from the Brazilian coast in the western South Atlantic. For fish larvae, pigmentation patterns are regarded as useful characters for taxonomic identification of closely related species. They are often established during early development, which is interesting when dealing with initial ontogenetic stages, but caution should be taken as some patterns are also highly convergent, with different groups presenting similar patterns (Fuiman et al., 1983; Moser et al., 1984; Moser, 1996; Richards, 2005; Miller & Kendall, 2009).

Moser (1996) described the larvae of two different species of Labrisomus (L. xanti and L. multiporosus) from the Pacific coast of California. The ventral pigmentation patterns described for the Pacific species are similar to the ones observed here for L. cricota and L. conditus. Within these pairs of species, one has fewer ventral spots with one enlarged spot in the mid-section of the body (L. conditus and L. xanti) while the other has more regularly sized and shaped spots, homogeneously distributed along the ventral region of the body of pre-flexion larvae (L. cricota and L. multiporosus). Similarities in the pigmentation pattern observed between species from different regions were not in accordance to phylogenetic proximity, where actually species from the same geographic region (L. cricota + L. conditus vs. L. xanti + L. multiporosus) are more closely related among themselves (Fig. 2). This suggests that evolution of the pigment pattern has followed a similar, apparently convergent, process during the speciation of both the Atlantic and Pacific species pairs, but further phylogenetic analyses including more species should be conducted to evaluate how spot patterns relate to the systematics and biogeography in Labrisomus. In sum, species relationships within Labrisomus have not been properly addressed yet as phylogenetic analyses performed so far did not include a broad ingroup species coverage, including our own presented here.

Regarding the larvae identified as Eucinostomus argenteus, the ventral spot pattern observed here might be common to other species of the genus as well. Moser (1996), for instance, was unable to anatomically distinguish among the three species that occur off California (E. argenteus, E. currani and E. entomelas). The ventral pigmentation pattern described by Moser (1996) is similar to the pattern observed in this study, with 6 to 8 spots on the ventral midline, compared to the 5 to 8 post-anal spots described for E. argenteus here and the 6 to 8 ventral midline spots reported for E. argenteus and E. gula by Richards (2005). The three pre-anal spots observed here correspond to the anterior and posterior pigmented region on the gut, plus one melanophore on the cleithral symphysis described by Moser (1996) and Richards (2005).

Conclusion

Our results encourage further use of molecular techniques for taxonomic identification of marine fish larvae in the western South Atlantic. More than 84% of the specimens were identified to species level (75 specimens), with only 14 specimens identified at the genus level probably due to the lack of proper reference sequences of the species. Therefore, we highlight that even for markers that have been used for more than a decade, more studies are needed to develop more encompassing reference sequence libraries, ideally with reliably identified voucher specimens deposited in fish collections. Ventral pigmentation is recommended as a possible diagnostic character among pre-flexion larvae of Labrisomus in the western South Atlantic, but additional specimens, including other species and localities across their geographic distributions, must be examined to test if these characters are reliable. Cryptobenthic reef fish species were predominant here, as observed in previous ichthyoplankton studies, and because of their relevance to reef and coastal island ecosystems, accurate species identification of those species is key for the conservation of these complex and biologically diverse environments.

Supplemental Information

Supplemental Information 1 K2P Intraspecific genetic distances.

Supplemental Information 2 K2P Interspecific genetic distances.

Supplemental Information 3 COI raw sequences.

We are especially grateful to boatman Paulo Sergio Peixoto de Moraes and biologist Thayna de Fatima Sarinho for all the help and support collecting ichthyoplankton.

Additional Information and Declarations

Competing Interests

Author Contributions

Field Study Permissions

Data Availability

Rodrigo Nunes-da-Fonseca is an Academic Editor for PeerJ.

Allan Pierre Bonetti Pozzobon conceived and designed the experiments, performed the experiments, analyzed the data, prepared figures and/or tables, authored or reviewed drafts of the article, and approved the final draft.

Jonathan Stuart Ready conceived and designed the experiments, analyzed the data, authored or reviewed drafts of the article, and approved the final draft.

Fabio Di Dario analyzed the data, authored or reviewed drafts of the article, and approved the final draft.

Rodrigo Nunes-da-Fonseca conceived and designed the experiments, analyzed the data, authored or reviewed drafts of the article, and approved the final draft.

The following information was supplied relating to field study approvals (i.e., approving body and any reference numbers):

Do not apply

The following information was supplied regarding data availability:

The Cytochrome Oxidase 1 (CO1) sequences are available in the Supplemental Files and at GenBank: PP578103 to PP578191.

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
