# Peer review of "Identification of pre-flexion fish larvae from the western South Atlantic using DNA barcoding and morphological characters"

_PeerJ, doi:10.7717/peerj.17791_

## Round 0.1 · original submission · Minor Revisions

Your manuscript has been reviewed by three external reviewers. The reviewers think that your work is of potential interest; however, they also raised several concerns about your study.

In particular, multiple reviewers have concerns about (1) the sampling being conducted in two consecutive months of the same season, and (2) the need to include intra- and inter-specific K2P distance in the results. Please address these in your revised manuscript.

Reviewer 1 ·

Basic reporting

Comments on the attached file

Experimental design

Comments on the attached file

Validity of the findings

Comments on the attached file

Annotated reviews are not available for download in order to protect the identity of reviewers who chose to remain anonymous.

·

Basic reporting

The manuscript investigates the use of DNA barcoding and morphological characters to identify pre-flexion fish larvae in the western South Atlantic. It focuses on the challenges of taxonomic identification due to the high diversity and similarity among early developmental stages of closely related fish species. By sequencing the mitochondrial COI-5P fragment from 89 fish larvae and comparing these with reference databases, the study identifies several species, mainly within the Blenniiformes order. Additionally, it explores anatomical characters, particularly ventral pigmentation patterns, as diagnostic tools for species identification among early stage larvae. Scientifically, this study enhances the accuracy of ichthyoplankton identification, crucial for effective conservation and management of marine biodiversity.
Summarize, this manuscript requires minor revisions.

Experimental design

no comment

Validity of the findings

'no comment

Additional comments

The manuscript investigates the use of DNA barcoding and morphological characters to identify pre-flexion fish larvae in the western South Atlantic. It focuses on the challenges of taxonomic identification due to the high diversity and similarity among early developmental stages of closely related fish species. By sequencing the mitochondrial COI-5P fragment from 89 fish larvae and comparing these with reference databases, the study identifies several species, mainly within the Blenniiformes order. Additionally, it explores anatomical characters, particularly ventral pigmentation patterns, as diagnostic tools for species identification among early stage larvae. Scientifically, this study enhances the accuracy of ichthyoplankton identification, crucial for effective conservation and management of marine biodiversity.
Summarize, this manuscript requires minor revisions.
General comments:
line 129, Figure 1, line 213, figure 2, should the first letter be capitalized? Please maintain consistency throughout the text.
The sampling detailed in the manuscript occurred over two consecutive months, March and April 2022, which suggests that it was conducted within a single season. This might limit the study's ability to capture seasonal variations in species diversity or reproductive patterns. pls, give more explanation;
The author successfully identified fish larvae through DNA barcoding based on data from NCBI; moreover, how can the relevance of the sequences on NCBI to this species be ensured? In other words, how can it be confirmed that these sequences indeed originate from this species and are not mislabeled? This is a common issue with molecular identification, and I would like to see the author's response.
The paper is somewhat stiff in its linguistic expression, suggesting a need for language polishing; further inspection of the reference formatting is also recommended.

Reviewer 3 ·

Basic reporting

Very relevant work for the knowledge of marine ichthyofauna. He applied the molecular tool quite appropriately. Clear and well-written text. However, to be considered for publication, it needs some repairs.
All suggestions for corrections and changes are marked in the commented text in Word, which is attached.

Experimental design

Well-delineated methods, however, need to better explain some aspects.
It needs to detail some important points, mainly regarding sample collection and processing.

Validity of the findings

Very valid and necessary data. However, it needs to better describe and show what was actually identified with the molecular tool.

Additional comments

Well-delineated and well-written work. With a good theoretical foundation and well developed. But some adjustments are necessary for publication.

Annotated reviews are not available for download in order to protect the identity of reviewers who chose to remain anonymous.

---

## Round 0.2 · Minor Revisions

While the reviewers and I fully understand that conducting a new sampling campaign in another season is not feasible nor essential for this work, I believe it would be good if the authors can at least acknowledge the fact that sampling was conducted in a single season as a limitation of the study in the discussion.

The authors have commented on reviewer 2's concern about the reliability of the public sequences. However, did the authors follow their own suggestions in this work? For instance, were any of the hits recovered from multiple sources or from voucher specimens? Again, the authors are recommended to discuss this aspect in the discussion.

Reviewer 3 ·

Basic reporting

Requested review was met

Experimental design

Requested review was met

Validity of the findings

Requested review was met

Additional comments

Requested review was met

---

## Round 0.3 · accepted · Accept

I confirm that you have addressed all of the reviewers' comments and am glad to inform you that this manuscript is now deemed ready for publication in PeerJ.